# Flanders Nursing Home (FLANH) project: Protocol of a multicenter longitudinal observational study on staffing, work environment, rationing of care, and resident and care worker outcomes

**Lisa Geyskens**[1,2], **Anja Declercq**[3,4], **Koen Milisen**[5], **Johan Flamaing**[1,6], **Mieke Deschodt**[1,7]*, **the FLANH research consortium**[¶]

1 Department of Public Health and Primary Care, Gerontology and Geriatrics, KU Leuven, Leuven, Belgium, 2 Research Foundation–Flanders (FWO), Brussels, Belgium, 3 LUCAS Centre for Care Research and Consultancy, KU Leuven, Leuven, Belgium, 4 CESO Centre for Sociological Research, KU Leuven, Leuven, Belgium, 5 Department of Public Health and Primary Care, Academic Centre for Nursing and Midwifery, KU Leuven, Leuven, Belgium, 6 Department of Geriatric Medicine, University Hospitals Leuven, Leuven, Belgium, 7 Competence Center of Nursing, University Hospitals Leuven, Leuven, Belgium

¶ Membership of the FLANH research consortium is provided in the Acknowledgments.
* mieke.deschodt@kuleuven.be

## Abstract

### Background

While the demand for high quality of care in nursing homes is rising, it is becoming increasingly difficult to recruit and retain qualified care workers. To date, evidence regarding key organizational factors such as staffing, work environment, and rationing of care, and their relationship with resident and care worker outcomes in nursing homes is still scarce. Therefore, the Flanders Nursing Home (FLANH) project aims to comprehensively examine these relationships in order to contribute to the scientific knowledge base needed for optimal quality of care and workforce planning in nursing homes.

### Methods

FLANH is a multicenter longitudinal observational study in Flemish nursing homes based on survey and registry data that will be collected in 2023 and 2025. Nursing home characteristics and staffing variables will be collected through a management survey, while work environment variables, rationing of care, and care worker characteristics and outcomes will be collected through a care worker survey. Resident characteristics and outcomes will be retrieved from the Belgian Resident Assessment Instrument for long-Term Care Facilities (BelRAI LTCF) database. Multilevel regression analyses will be applied to examine the relationships between staffing variables, work environment variables, and rationing of care and resident and care worker outcomes.

**Data Availability Statement:** No datasets were generated or analyzed during the current study. All relevant data from this study will be made available upon study completion.

**Funding:** LG is supported by a PhD fellowship of the Research Foundation – Flanders (11O6123N). The FLANH project is funded by KU Leuven Internal Funds (3M220304). The funders did not and will not have a role in study design, data collection and analysis, decision to publish, or preparation of the manuscript.

**Competing interests:** The authors have declared that no competing interests exist.

## Conclusion

This study will contribute to a comprehensive understanding of the nursing home context and the interrelated factors influencing residents and care workers. The findings will inform the decision-making of nursing home managers and policymakers, and evidence-based strategies to optimize quality of care and workforce planning in nursing homes.

## Introduction

Over the past decades, the nursing home population has become older and more frail, and their care needs have become more complex [1]. Nursing home care workers are therefore increasingly challenged to provide high quality of care. At the same time, the number of care workers and their skill level have remained relatively static due to rising healthcare expenditures and difficulties in recruiting and retaining qualified professionals [2]. The COVID-19 pandemic has made it even clearer how the growing gap between increasing demand and limited resources threatens the quality of care and the maintenance of a healthy workforce in nursing homes [3].

Quality of care and its relationship with staffing variables (i.e. staffing level, skill mix, and turnover) has been studied extensively, and while there is a tendency for better staffing variables to be associated with better resident outcomes, the evidence remains inconclusive [4–6]. This lack of concrete findings may be related to methodological limitations, such as the fact that most studies are cross-sectional or only examine a single staffing variable instead of combining them. For instance, the staffing level of a nursing home may be high, but if the skill level of the care workers is insufficient to meet the complex care needs of residents, the quality of care may still be poor. Furthermore, in the hospital setting, it has been suggested that staffing improvements alone may have limited impact on quality of care without a good work environment in place [7]. Indeed, in the nursing home setting, there is a growing body of literature showing that work environment variables, such as supportive leadership, good safety climate, and lower workload are related to better resident outcomes, as well as better care worker outcomes [8–14]. But again, most studies are cross-sectional and usually focus on single elements of the work environment. Another variable to consider is rationing of care, as this might play an important mediating role between the relationships of staffing and work environment variables and resident and care worker outcomes. Rationing of care is the withholding of or failure to carry out necessary care due to a lack of resources (e.g. inadequate time, staffing levels, and/or skill mix) [15]. Although this phenomenon has rarely been studied in nursing homes, evidence suggests that poor staffing and work environment variables can lead to rationing of care [16–19], which in turn can negatively impact both resident and care workers outcomes [20–23].

To date, only a few studies, all with a cross-sectional design, have combined staffing and work environment variables when examining resident or care worker outcomes in nursing homes [10–14], and of these, only one has integrated rationing of care [10]. This protocol paper will describe the Flanders Nursing Home (FLANH) project, a study that aims to comprehensively examine the interrelationships of these variables and outcomes using a longitudinal design. We hypothesize that better staffing and work environment variables and lower levels of rationing of care are related to better resident and care worker outcomes.

### Nursing home context in Flanders

Nursing homes in Flanders, the Dutch-speaking region of Belgium, are facilities where care and support is provided to care-dependent older people who reside there permanently, and

where residents receive additional medical care from their own general practitioner visiting the nursing home [24]. Each nursing home also has an appointed coordinating and advising physician who has a connecting role between the nursing home staff and the group of attending general practitioners [24]. Anno 2023, Flanders has a total of 826 nursing homes that vary in size and ownership status [25]. In 2020, 5.3% of the Flemish population aged 65 years and older lived in a nursing home, and for men and women aged 85 years and older, this was 14% and 26% respectively [26]. Over the years, their care burden profile has systematically increased from 70% of residents being severely care-dependent in 2010 to 82% in 2020 [26]. To optimize the quality and continuity of care, the Flemish government has imposed the use of the Belgian Resident Assessment Instrument for Long-Term Care Facilities (BelRAI LTCF) in all nursing homes from June 2023 [24]. The BelRAI LTCF is the Belgian version of the internationally validated interRAI LTCF instrument, which is an assessment tool to evaluate a resident's physical, social and psychological functioning and care needs [27].

## Aim and objectives

The overall aim of FLANH is to develop a comprehensive understanding of the relationships between staffing variables, work environment variables, and rationing of care and resident and care worker outcomes in Flemish nursing homes.

The specific objectives are:

1. To describe staffing variables, work environment variables, rationing of care and resident and care worker outcomes.

2. To examine the cross-sectional relationships between staffing variables, work environment variables, and rationing of care and resident and care worker outcomes.

3. To examine if staffing variables, work environment variables, and rationing of care can predict resident and care worker outcomes over time.

4. To examine the impact of change in staffing variables, work environment variables, and rationing of care on resident and care worker outcomes over time.

## Methods

### Design

FLANH is a multicenter longitudinal observational study in Flemish nursing homes based on survey and BelRAI LTCF data that will be collected at two timepoints (2023 and 2025).

### Setting and sample

In 2023, all 120 nursing homes for which BelRAI LTCF data can be retrieved from a central database will be invited to participate in this study, which currently corresponds to 14.5% of all Flemish nursing homes. This sample may be expanded in 2025, as by then all nursing homes are expected to be actively using the BelRAI LTCF for all their residents.

In each participating nursing home, a management representative and all care workers who understand Dutch and provide direct care to residents will be surveyed. Care workers on long-term leave (>1 month), temporary employees, students, and volunteers will be excluded. Data of all residents living in the participating nursing homes will be collected from the BelRAI LTCF database.

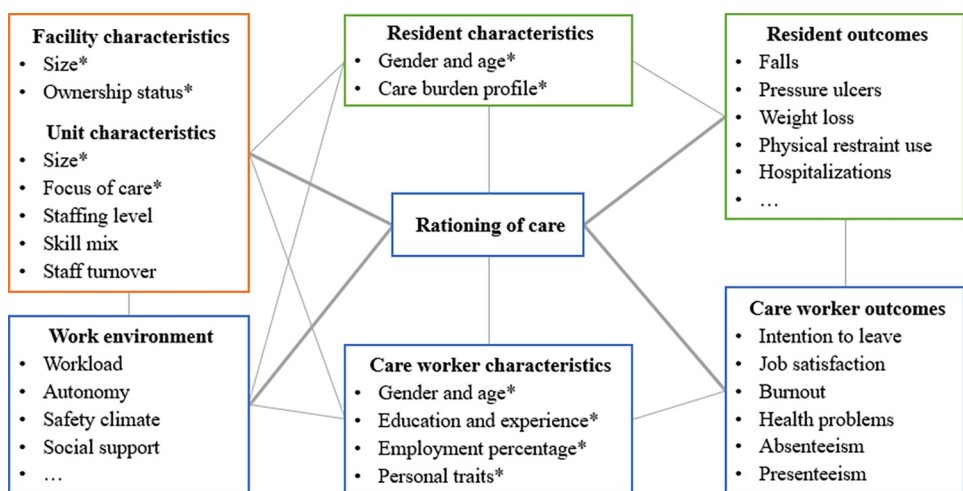

Variables from management survey, variables from care worker survey, variables from BelRAI LTCF database

**Fig 1. Overview of the survey and BelRAI LTCF variables (adapted from SHURP [10]).** *Control variables that will be included in the analyses.

## Data sources, variables and measurements

Three data sources will be used: (1) a management survey to assess facility and unit characteristics, including staffing variables; (2) a care worker survey to assess work environment variables, rationing of care, and care worker characteristics and outcomes; and (3) the BelRAI LTCF database to obtain resident characteristics and outcomes. The variables and outcomes of interest (Fig 1) are largely based on the nursing home quality of care framework of the Swiss Nursing Homes Human Resources Project (SHURP) [10].

**Survey development.** To develop the management and care worker survey, we used the existing SHURP questionnaire [10] as a starting point and reviewed the literature regarding staffing calculation methods, key aspects of the work environment and care worker well-being, rationing of care, quality of care indicators, etc. We also consulted the External Service for Prevention and Protection at Work, a Belgian organization with expertise in psychosocial risk assessments of work environments. Based on this literature review and expert advice, we selected established scales and items to assess the variables and outcomes of interest. Where necessary, adaptations were made for use in the nursing home setting (e.g. referring to 'resident' instead of 'patient') and for scales not available in Dutch, a forward translation was performed by a bilingual researcher. Subsequently, the preliminary surveys were reviewed by a group of experts in residential care including researchers and policy officers with diverse backgrounds in nursing, sociology, psychology, and healthcare management. Based on their input, it was decided to omit certain items in order to reduce the survey burden. Also, items with complicated phrasing were adapted to enhance comprehensibility for care workers of all educational levels. Finally, the management survey was pretested by a nursing home director, and the care worker survey was pretested through a focus group interview using open discussion with care workers from various professions (i.e. two registered nurses, four care assistants, one occupational therapist, one logistic worker). After incorporating their suggestions, the surveys were finalized.

**Management survey.** We will use the management survey to assess facility characteristics at the nursing home level and unit characteristics. The facility characteristics include nursing home size (i.e. number of units, number of beds), ownership status (i.e. public, private for-

profit, private non-profit), format of resident records (i.e. paper or electronic), quality of the collaboration with the coordinating and advising physician as perceived by management, number of care workers per profession type (e.g. registered nurses, nurse assistants, allied health professionals, logistic workers, social workers, animation staff, etc.), and number of full time equivalents per profession type.

The unit characteristics include unit size (i.e. number of residents living on the unit), focus of care (e.g. dementia, psychiatric), number of care workers working specifically on that unit, number of care workers from that unit who left the nursing home in the past six months, and an actual work roster of that unit of an average day including number of care workers that worked that day, their profession type and the start and finish time of their shift. Staffing level will be assessed using hours per resident day (HPRD). This will be calculated by dividing the total number of productive hours of all care workers with direct care responsibilities on that unit by the number of residents living on that unit [28]. Productive hours are the actual worked hours and do not include paid or scheduled hours for vacation, sick leave, or education [28]. Skill mix will be calculated as the proportions of total productive hours by each profession type [28]. Staff turnover will be calculated by dividing the total number of care workers who left the nursing home in the past six months by the number of care workers employed at the time of the survey.

**Care worker survey.** We will use the care worker survey to assess the work environment, rationing of care, and care worker characteristics and outcomes. An overview of the main variables and their measurements is provided in Table 1. In addition, the following care worker characteristics will be collected: gender, age, type of profession, educational level, country of education, employment percentage, usual work shift (i.e. day shifts, night shift, regular change between day and night shifts), number of years of professional experience in the nursing home, and number of years of professional experience as a care worker in general.

**BelRAI LTCF database.** We will retrieve resident characteristics and outcomes form the BelRAI LTCF database. BelRAI LTCF data are routinely collected, in electronic format, by nursing home care workers who are trained BelRAI assessors. The instrument consists of 18 sections, including demographics, cognition, mood, behavior, functional status, health conditions, medications, social activities, advance directives, etc.

The resident characteristics that will be retrieved are gender, age, care burden profile (i.e. functional and/or cognitive dependency level), length of stay, and mortality. The main resident outcomes that will be used are falls, pressure ulcers (i.e. stage 2 or higher), weight loss (i.e. 5% or more in the past month or 10% or more in the past 6 months), and use of physical restraints (i.e. bedrails, trunk restraint or chair that prevents rising). Additional resident outcomes captured with the BelRAI LTCF may also be used, such as hospitalization or emergency department visit, urinary tract infection, oral health, behavioral symptoms, self-reported mood, self-perceived health, social interaction, etc.

## Data collection and ethical considerations

First, an email will be sent to the selected nursing homes with an information letter and an invitation to an online meeting where the study purpose and procedures will be explained in detail. Subsequently, a management representative of each nursing home willing to participate will be asked to sign a written informed consent form and complete the management survey. Second, we will visit each participating nursing home to discuss the staffing data from the management survey and to make practical arrangements on the care worker data collection (e.g. planning, communication, supporting high response rate). Care workers will be able to access the online care worker survey through an URL or QR-code, which will be distributed by

**Table 1. Overview of care worker survey variables and their measurements.**

| Variable | Measurement, (answer options; interpretation) |
|---|---|
| *Work environment* | |
| Perceived staffing adequacy | 3 items from the subscale 'Staffing and Resource Adequacy' of the 'Practice Environment Scale of the Nursing Work Index' (PES-NWI) [29], (1 = strongly disagree to 5 = strongly agree; higher values indicating better perceived staffing adequacy) |
| Workload | 3-item subscale from the 'Short Inventory to Monitor Psychosocial Hazards' (SIMPH) [30], (1 = strongly disagree to 5 = strongly agree; higher values indicating higher workload) |
| Emotional burden | 3-item subscale from the SIMPH [30], (1 = strongly disagree to 5 = strongly agree; higher values indicating higher emotional burden) |
| Role clarity | 3-item subscale from the SIMPH [30], (1 = strongly disagree to 5 = strongly agree; higher values indicating better role clarity) |
| Skill use | 3-item subscale from the SIMPH [30], (1 = strongly disagree to 5 = strongly agree; higher values indicating better skill use) |
| Training opportunities | 3 items investigator-developed, (1 = strongly disagree to 5 = strongly agree; higher values indicating better training opportunities) |
| Work-life balance | 3-item from the 'Interrole conflict' scale [31], (1 = strongly disagree to 5 = strongly agree; higher values indicating poorer work-life balance) |
| Autonomy | 4 items from the 'Autonomy and Control scale' [32] and 1 item from the de 'Nursing Work Index-Revised' (NWI-R) [33], (1 = strongly disagree to 5 = strongly agree; higher values indicating more autonomy) |
| Salary | 1 item from the subscale 'Payment' of the 'Questionnaire on Perception and Judgement of Work' [34], (1 = strongly disagree to 5 = strongly agree; higher values indicating greater satisfaction with salary) |
| Involvement | 3 items form the subscale 'Involvement' of the 'Organizational Climate Measure' (OCM) [35], (1 = strongly disagree to 5 = strongly agree; higher values indicating greater involvement) |
| Safety climate | 4 items from the subscale 'Safety climate' of the 'Safety Attitudes Questionnaire' (SAQ) [36], (1 = strongly disagree to 5 = strongly agree; higher values indicating better safety climate) |
| Person-centered vision | 2 items from the subscale 'Extent of personalizing care' of 'the Person-centered Care Assessment Tool' (P-CAT) [37] and the 3-item subscale 'Patient and next of kin focus' of the 'Brisbane Practice Environment Measure for Nursing Homes' (B-PEM-NH) [38], (1 = strongly disagree to 5 = strongly agree; higher values indicating greater person-centered vision) |
| Collaboration with coordinating and advising physician | 1 item from the subscale 'Collegial Nurse–Physician Relations' of the PES-NWI [29], (1 = strongly disagree to 5 = strongly agree; higher values indicating better collaboration) |
| Social support of colleagues | 4-item adapted version of the 'Colleagues Support' scale [39], (1 = strongly disagree to 5 = strongly agree; higher values indicating better social support of colleagues) |

(*Continued*)

**Table 1.** (Continued)

| Variable | Measurement, (answer options; interpretation) |
|---|---|
| Social support of supervisor | 4-item adapted version of the 'Supervisor Support' scale [39], (1 = strongly disagree to 5 = strongly agree; higher values indicating better social support of supervisor) |
| *Care and support* | |
| Rationing of care | 20-item adapted version of the 'Basel Extent of Rationing of Nursing Care for Nursing Homes instrument' (BERNCA-NH) [40], (1 = never to 4 = often with option 0 = not applicable; higher values indicating higher levels of rationing of care) |
| Perceived quality of care | Single-item rating of overall quality of care [41], (1–10 rating scale; higher values indicating better quality of care) |
| *Care worker outcomes* | |
| Burnout | 12-item short version of the 'Burnout Assessment Tool' (BAT) [42], (1 = never to 5 = always; higher values indicating higher levels of burnout) |
| Physical complaints (i.e. back pain, joint pain, tiredness, sleeplessness, headache, work-related allergies) | 6 items from the SHURP questionnaire [10], (1 = never to 5 = daily; higher values indicating more frequent physical complaints) |
| Intention to leave | 3-item 'Turnover Intention' scale [43], (1 = strongly disagree to 5 = strongly agree; higher values indicating greater intention to leave) |
| Job satisfaction | Single-item rating of overall job satisfaction [44], (1–10 rating scale; higher values indicating greater job satisfaction) |
| Absenteeism (i.e. unplanned absence due to illness) | 1 item investigator-developed, (0 = no times to 5 = five times or more; higher values indicating higher absenteeism) |
| Presenteeism (i.e. going to work while being ill) | 1 item investigator-developed, (0 = no times to 5 = five times or more; higher values indicating higher presenteeism) |
| *Personal traits* | |
| Self-efficacy | 3 items from the short version of the 'Occupational Self-Efficacy Scale' (OSES) [45], (1 = strongly disagree to 5 = strongly agree, with higher values indicating higher levels of self-efficacy) |
| Resilience | 3 items from the 'Brief Resilience Scale' (BRS) [46], (1 = strongly disagree to 5 = strongly agree; higher values indicating higher levels of resilience) |
| Optimism | 3-item subscale 'Optimism' of the 'Life Orientation Test-Revised' (LOT-R) [47], (1 = strongly disagree to 5 = strongly agree; higher values indicating higher levels of optimism) |

the management representative to all eligible care workers of the nursing home. An information letter and electronic informed consent form will be provided online describing the study purpose, data protection and voluntary nature of participation. After their consent, care workers can start completing the online survey. The baseline data collection will take place from February-July 2023. Two years later, the data collection process will be repeated among all participating nursing homes. We will use REDCap, a secure web application, to collect, manage and store all care worker survey data. When the data will be exported from REDCap, identifying information will be removed to pseudonymize the data. Finally, pseudonymized data of all residents living in the participating nursing homes will be retrieved from the BelRAI LTCF database at baseline and follow-up. Resident data will then be matched to management and care worker data at the nursing home unit level using unit-specific identifiers. All data will be stored on a password-protected server accessible only to authorized members of the research team (LG and MD).

## Data analysis

In accordance with the study objectives, data analysis will be performed as follows:

- Objective 1: Descriptive statistics (i.e. frequencies, percentages, means and standard deviations, median and interquartile ranges) will be calculated as appropriate to describe the variables and outcomes of interest.

- Objective 2: To examine the cross-sectional relationships between staffing variables, work environment variables, and rationing of care and resident and care worker outcomes, we will use a three-level regression analysis. Multilevel modelling bases on mixed models [48, 49] will account for the clustering of residents and care workers (level 1) within nursing home units (level 2) within nursing home facilities (level 3). Staffing variables, work environment variables, and rationing of care will be analyzed at the unit level and resident and care worker outcomes at the individual level.

- Objective 3: To identify predictors, we will examine the relationships between staffing variables, work environment variables, and rationing of care measured at baseline and resident and care worker outcomes at follow-up.

- Objective 4: We will examine the relationships between changes in staffing variables, work environment variables, and rationing of care and resident and care worker outcomes over time. Changes will be estimated between baseline and follow-up.

Incomplete data will be assumed to be missing at random (MAR; meaning that missingness is allowed to depend on observed covariates and outcomes, but not on unobserved information). This allows for ignorable likelihood analysis and, with incomplete covariates, multiple imputation [50]. The MAR assumption will be challenged in a sensitivity analysis.

## Stakeholder involvement

Since the proposal writing phase of the FLANH project, we are closely collaborating with a large stakeholder group, including representatives of the three umbrella organizations that defend the interests of nursing homes in Flanders, namely Zorgnet-Icuro, the Flemish Independent Care Network (VLOZO), and the Association of Flemish Cities and Municipalities (VVSG). Also represented in the stakeholder group are the following organizations: the Agency for Care and Health of the Flemish government, the Flemish Institute for Quality of Care (VIKZ), the Flemish Council of Older People which is the official platform for political participation of older people in the Flemish government's elderly policy, the software company Pyxima that led the development of the BelRAI platform for the federal government, the External Service for Prevention and Protection at Work (IDEWE) which aims to improve working environments and psychosocial well-being of employees, and the Belgian Society for Gerontology and Geriatrics (BSGG) which represents gerontological healthcare workers and promotes scientific research in the field.

The stakeholders are consulted on a regular basis to exchange ideas and feedback on ongoing research activities (e.g. survey design, development of recruitment strategies, communication plan). They will also support the research team in formulating policy recommendations based on the results, as well as outreach activities to the public (e.g. dissemination of findings, public talks, website, lay publications, learning network, final project report in Dutch).

## Recruitment strategies

To raise awareness of the project and create a sense of urgency and necessity to participate, FLANH will be highlighted through various communication channels, such as newsletters and

websites of the stakeholder organizations, social media posts and talks at relevant events. Furthermore, multiple incentives will be provided to encourage participation. First, as an added value for the participating nursing homes, they will each receive a benchmarking report with their descriptive results. This will help nursing home managers identify the strengths and weaknesses of their facility and use them as a basis for future improvements initiatives. Second, participating nursing homes, including care workers, will have the opportunity to join a learning network of FLANH. Through this network, we will offer free training on relevant topics and create a platform where practical knowledge and experiences can be exchanged. Third, movie tickets will be raffled off among care workers who completed the survey. Lastly, as a final incentive, we will reward all care workers of nursing homes where a minimum response rate of 75% is achieved with cake.

## Discussion

While research on quality of care and workforce planning in the hospital setting has been thriving in the past decades [51–53], research in the nursing home setting has not received the attention it deserves. This has only recently changed with the COVID-19 pandemic, which exposed and exacerbated the many challenges in nursing homes that have likely existed for years. In response to these scientific gaps and societal challenges, this study will generate the necessary scientific knowledge base that is currently lacking to underpin the decision-making of nursing home managers and policymakers and to inform future quality improvement projects and more effective strategies to support recruitment and retention of qualified care workers in nursing homes.

FLANH will be the first study in the Flemish nursing home context to comprehensively examine the relationships between staffing variables, work environment variables, and rationing of care with a dual focus on both resident and care workers outcomes. To analyze these relationships, we will look at all variables simultaneously and use a multilevel model to account for mediating and moderating factors. This will allow to more clearly distinguish at which level and in which areas specific interventions should be targeted to improve residents and care worker outcomes. Moreover, the longitudinal design will also allow analyses of the predictive value and changes over time of the study variables. This will be unique in Flanders, but will also be a landmark internationally, as the bulk of available evidence is based on cross-sectional data analyses. A potential risk of this study, however, is a low participation rate. Currently, nursing homes and care workers are under great pressure, and while this may be a reason for some to participate, it is more likely to be a hindrance. To address this, we will use a multi-channel communication strategy to raise awareness of the project and provide incentives to encourage participation. Nevertheless, we are aware that this study will be susceptible to selection bias. Therefore, the facility characteristics of the Flemish nursing homes that do and do not participate will be compared and this will be reported transparently. Another potential risk to consider is response bias, as care workers may tend to give responses influenced by social desirability. To mitigate this, we will emphasize that all surveys will be sent directly to the research team and that confidentiality will be guaranteed.

## Supporting information

**S1 Checklist. STROBE statement—checklist of items that should be included in reports of observational studies.**
(DOCX)

## Acknowledgments

Members of the FLANH research consortium: Mieke Deschodt (Gerontology and Geriatrics, KU Leuven), Anja Declercq (LUCAS Centre for Care Research and Consultancy, KU Leuven), Koen Milisen (Academic Centre for Nursing and Midwifery, KU Leuven), Johan Flamaing (Gerontology and Geriatrics, KU Leuven), Lisa Geyskens (Gerontology and Geriatrics, KU Leuven), Pieter Heeren (Academic Centre for Nursing and Midwifery, KU Leuven), Ellen Vlaeyen (Faculty of Medicine and Life Sciences, Hasselt University; Academic Centre for Nursing and Midwifery, KU Leuven), Kris Vanhaecht (Leuven Institute for Healthcare Policy, KU Leuven), Gijs Van Pottelbergh (Academic Center for General Practice, KU Leuven), Lode Godderis (Environment and Health, KU Leuven; External Service for Prevention and Protection at Work, IDEWE), Jeroen Trybou (Healthcare Management and Policy, UGent), Jan Hamers (Health Services Research, Maastricht University), Ramona Backhaus (Health Services Research, Maastricht University), and Franziska Zúñiga (Institute of Nursing Science, University of Basel).

We would like to thank Geert Molenberghs (Leuven Biostatistics and Statistical Bioinformatics Centre, KU Leuven) who contributed to the development of the statistical approach and Eva Guldentops, project manager of the FLANH project who is coordinating the data collection of the 2023 sample and is responsible for the communication with the nursing homes. We also thank the members of the stakeholder group for their valuable input and support.

## Author Contributions

**Conceptualization:** Lisa Geyskens, Anja Declercq, Koen Milisen, Johan Flamaing, Mieke Deschodt.

**Funding acquisition:** Lisa Geyskens, Johan Flamaing, Mieke Deschodt.

**Methodology:** Lisa Geyskens, Anja Declercq, Koen Milisen, Mieke Deschodt.

**Project administration:** Lisa Geyskens, Mieke Deschodt.

**Supervision:** Mieke Deschodt.

**Writing – original draft:** Lisa Geyskens, Mieke Deschodt.

**Writing – review & editing:** Lisa Geyskens, Anja Declercq, Koen Milisen, Johan Flamaing, Mieke Deschodt.

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
