## [Decision Letter · Decision Letter 0]

30 Aug 2023

PONE-D-23-16826Flanders Nursing Home (FLANH) project: Protocol of a multicenter longitudinal observational study on staffing, work environment, rationing of care, and resident and care worker outcomes

PLOS ONE

Dear Dr. Deschodt,

Thank you for submitting your manuscript to PLOS ONE. After careful consideration, we feel that it has merit but does not fully meet PLOS ONE’s publication criteria as it currently stands. Therefore, we invite you to submit a revised version of the manuscript that addresses the points raised during the review process.

We look forward to receiving your revised manuscript.

Kind regards,

Jonas Preposi Cruz

Academic Editor

PLOS ONE

2. One of the noted authors is a group or consortium [Leuven Institute for Healthcare Policy, KU Leuven]. In addition to naming the author group, please list the individual authors and affiliations within this group in the acknowledgments section of your manuscript. Please also indicate clearly a lead author for this group along with a contact email address.

Reviewers' comments:

Reviewer's Responses to Questions

**Comments to the Author**

1. Does the manuscript provide a valid rationale for the proposed study, with clearly identified and justified research questions?

Reviewer #1: Yes

Reviewer #2: Yes

2. Is the protocol technically sound and planned in a manner that will lead to a meaningful outcome and allow testing the stated hypotheses?

Reviewer #1: Yes

Reviewer #2: Yes

3. Is the methodology feasible and described in sufficient detail to allow the work to be replicable?

Reviewer #1: Yes

Reviewer #2: Yes

4. Have the authors described where all data underlying the findings will be made available when the study is complete?

Reviewer #1: Yes

Reviewer #2: Yes

5. Is the manuscript presented in an intelligible fashion and written in standard English?

Reviewer #1: Yes

Reviewer #2: Yes

6. Review Comments to the Author

You may also provide optional suggestions and comments to authors that they might find helpful in planning their study.

Reviewer #1: This is a longitudinal study protocol that aims to investivate Flemish Nursing Homes' staffing, work environment, rationing of care and resident and care worker outcomes. The study is relevant in that it contributes to understanding on human resource and work environment and its impact to several outcomes. In particular, this study will provide evidence in the context of Nursing Homes.

However, the following are important review comments that can further improve the paper:

1. In Data Sources, particularly, the survey development, it was mentioned in lines145 - 146 that the scales were translated if not available in dutch. Provide the measured done or planned to ensure validity of the translated scales.

2. In the Data Analysis, describe the analysis procedures and parameters in the interpretation of the included scales.

3. In the discussion section, strengths and limitations were explained. Limitations were backed up by planned measures to mitigate its effects. The authors might also want to consider adding the use of self-report questionnaires as a limitation and possible source of bias. Probable, the authors would like to add measures to address these limitations.

Reviewer #2: Dear Authors,

the protocol of your study is well presented.

Some minor suggestions:

in line 172 explain what is understood by "productive hours"

section from 186-200: are the quality indicators you use from the flemish Institute for Quality of Care retrievable from the BelRAI databse? If not, think about to give the section another title.

7. PLOS authors have the option to publish the peer review history of their article (what does this mean?). If published, this will include your full peer review and any attached files.

Reviewer #1: **Yes: **Joel Estacio

Reviewer #2: No

---

## [Author Response · Author response to Decision Letter 0]

22 Sep 2023

COMMENTS EDITOR

2. One of the noted authors is a group or consortium. In addition to naming the author group, please list the individual authors and affiliations within this group in the acknowledgments section of your manuscript. Please also indicate clearly a lead author for this group along with a contact email address.

We would like to refer to the Acknowledgments section (lines 310-318) where all members of the FLANH research consortium are listed. The authors of this manuscript are actually also part of the consortium and are hereby also added. The lead author of the consortium is prof. Mieke Deschodt, who is the corresponding author of this manuscript. Her contact email address is provided on the title page, and again added in the Acknowledgments section.

No changes were made to the reference list.

COMMENTS REVIEWER #1

1. In Data Sources, particularly, the survey development, it was mentioned in lines145 - 146 that the scales were translated if not available in Dutch. Provide the measured done or planned to ensure validity of the translated scales.

Most of the scales and items were available in Dutch, but some indeed had to be translated. A bilingual researcher who was fluent in both English and Dutch performed a forward translation, which was then reviewed by a second bilingual researcher. In addition, as we describe in the manuscript, an expert panel review was performed were linguistic and conceptual equivalence was assessed. We also conducted a pre-test with end-users to identify any potential issues with comprehension or wording. All feedback was used to refine the translations further.

However, although we believe appropriate steps were taken to ensure the reliability of the translations, for pragmatic reasons no backward translation was performed. We acknowledge that this is a limitation and have therefore reformulated the sentence to make it clear that only forward translation was performed (lines 145-148): “Where necessary, adaptations were made for use in the nursing home setting (e.g. referring to ‘resident’ instead of ‘patient’) and for scales not available in Dutch, a forward translation was performed by a bilingual researcher.”

2. In the Data Analysis, describe the analysis procedures and parameters in the interpretation of the included scales.

Thank you for this suggestion. To provide more clarity, we have described the anchor of answer options and score interpretation for each included scale in Table 1. In the following results papers, we will of course go into more detail regarding scale calculation and transformation for statistical analysis.

3. In the discussion section, strengths and limitations were explained. Limitations were backed up by planned measures to mitigate its effects. The authors might also want to consider adding the use of self-report questionnaires as a limitation and possible source of bias. Probable, the authors would like to add measures to address these limitations.

Thank you for this suggestion. We acknowledge the importance of addressing this limitation and have added the following in the discussion section (lines 305-308): “Another potential risk to consider is response bias, as care workers may tend to give responses influenced by social desirability. To mitigate this, we will emphasize that all surveys will be sent directly to the research team and that confidentiality will be guaranteed.”

COMMENTS REVIEWER #2

1. In line 172 explain what is understood by "productive hours"

Thank you for this comment. We have added the following sentence to clarify this (lines 173-174): “Productive hours are the actual worked hours and do not include paid or scheduled hours for vacation, sick leave, or education.”

2. Section from 186-200: are the quality indicators you use from the Flemish Institute for Quality of Care retrievable from the BelRAI database? If not, think about to give the section another title.

Currently, the Flemish Institute for Quality of Care gathers the quality indicators at the nursing home level through self-reporting by nursing home management and not through the BelRAI. However, as part of the BelRAI assessment, these quality indicators and many more outcomes are now also being collected at the resident level. Although the Flemish Institute for Quality of Care is planning to switch to the BelRAI in the future, we already decided to retrieve these outcomes from the BelRAI database.

So to answer the question, all resident outcomes we will use will be retrieved from the BelRAI database and will not be obtained from the Flemish Institute for Quality of Care. We agree that the text reads ambiguously and have therefore removed the reference to the Flemish Institute for Quality of Care from the paragraph (lines 195-198).

---

## [Decision Letter · Decision Letter 1]

17 Oct 2023

Flanders Nursing Home (FLANH) project: Protocol of a multicenter longitudinal observational study on staffing, work environment, rationing of care, and resident and care worker outcomes

PONE-D-23-16826R1

Dear Dr. Deschodt,

We’re pleased to inform you that your manuscript has been judged scientifically suitable for publication and will be formally accepted for publication once it meets all outstanding technical requirements.

Kind regards,

Jonas Preposi Cruz

Academic Editor

PLOS ONE

Additional Editor Comments (optional):

Reviewers' comments:

Reviewer's Responses to Questions

**Comments to the Author**

1. Does the manuscript provide a valid rationale for the proposed study, with clearly identified and justified research questions?

Reviewer #1: Yes

2. Is the protocol technically sound and planned in a manner that will lead to a meaningful outcome and allow testing the stated hypotheses?

Reviewer #1: Yes

3. Is the methodology feasible and described in sufficient detail to allow the work to be replicable?

Reviewer #1: Yes

4. Have the authors described where all data underlying the findings will be made available when the study is complete?

Reviewer #1: Yes

5. Is the manuscript presented in an intelligible fashion and written in standard English?

Reviewer #1: Yes

6. Review Comments to the Author

You may also provide optional suggestions and comments to authors that they might find helpful in planning their study.

Reviewer #1: The article is a protocol. Ensure that all ethical considerations are met during the implementation. Provide descriptions of ethical considerations in the publication of results.

7. PLOS authors have the option to publish the peer review history of their article (what does this mean?). If published, this will include your full peer review and any attached files.

Reviewer #1: **Yes: **Joel Estacio

---

## [Editor Report · Acceptance letter]

19 Oct 2023

PONE-D-23-16826R1 

Flanders Nursing Home (FLANH) project: Protocol of a multicenter longitudinal observational study on staffing, work environment, rationing of care, and resident and care worker outcomes 

Dear Dr. Deschodt:

I'm pleased to inform you that your manuscript has been deemed suitable for publication in PLOS ONE. Congratulations! Your manuscript is now with our production department. 

Kind regards, 

on behalf of

Dr. Jonas Preposi Cruz 

Academic Editor

PLOS ONE